# Improved Progression-Free Long-Term Survival of a Nation-Wide Patient Population with Metastatic Melanoma

**DOI:** 10.3390/cancers12092591

**Published:** 2020-09-11

**Authors:** Anne Vest Soerensen, Eva Ellebaek, Lars Bastholt, Henrik Schmidt, Marco Donia, Inge Marie Svane

**Affiliations:** 1Department of Oncology, Herlev Hospital, Copenhagen University Hospital, Borgmester Ib Juuls Vej 1, 2730 Herlev, Denmark; eva.ellebaek.steensgaard@regionh.dk (E.E.); Marco.Donia@regionh.dk (M.D.); Inge.Marie.Svane@regionh.dk (I.M.S.); 2Department of Oncology, Odense University Hospital, J.B. Winsloews vej 4, 5000 Odense C, Denmark; Lars.Bastholt@rsyd.dk; 3Department of Oncology, Aarhus University Hospital, Palle Juul Jensens Boulevard 99, Indgang C, Plan 6, Krydspunkt 605, 8200 Aarhus N, Denmark; henrschm@rm.dk; 4National Center for Cancer Immune Therapy, Department of Oncology, Herlev Hospital, Copenhagen University Hospital, Borgmester Ib Juuls Vej 25C, 2730 Herlev, Denmark

**Keywords:** long-term survival, nation-wide, real-life, metastatic melanoma, immune checkpoint-inhibitor

## Abstract

**Simple Summary:**

New cancer treatments have prolonged the lives for patients with metastatic melanoma in clinical trials. However, patients in clinical trials often presents with a better prognosis than other patients. It is therefore important to examine the effect of the new treatments in a real-life setting. We found that survival for patients with metastatic melanoma in a nation-wide setting were prolonged and a higher proportion of these patients were alive and without active disease 3 years after they started treatment.

**Abstract:**

Approval of immune checkpoint-inhibitors (ICIs) and BRAF-inhibitors has revolutionized the treatment of metastatic melanoma. Although these drugs have improved overall survival (OS) in clinical trials, real-world evidence for improved long-term survival is still scarce. Clinical data were extracted from the Danish Metastatic Melanoma database. This nation-wide cohort contains data on all patients who received systemic treatment for metastatic melanoma between 2008 and 2016. Ipilimumab, the first approved ICI, was implemented as standard-of-care in Denmark in 2012. Hence, patients were divided in a pre-ICI (2008–2011) and an ICI (2012–2016) era. Patients were defined as long-term survivors if they were alive 3 years after initiation of systemic therapy. Data from 1754 patients were retrieved. Patients treated in the ICI era had an improved median OS (11.3 months, 95% confidence interval (CI) 10.3–12.3) compared with those in the pre-ICI era (median OS 8.3 months, 95% CI 7.4–9.5, *p* < 0.0001). A higher proportion of long-term survivors was observed in the ICI era (survivors >3 years increased from 13% to 26% and survivors >5 years increased from 9% to 21%; both *p* < 0.0001). For long-term survivors, known prognostic factors were equally distributed between the two periods, except that long-term survivors in the pre-ICI era were younger. For long-term survivors, 70% were without progression in the ICI era compared with 43% in the pre-ICI era (*p* < 0.0001). For all patients, the proportion without progression increased from 5% to 18% between the pre-ICI and the ICI era (*p* < 0.0001), respectively. Implementation of ICI has led to a significant increase in progression-free, long-term survival for real-life patients with metastatic melanoma.

## 1. Introduction

The implementation of immune checkpoint-inhibitors (ICIs) and BRAF (v-raf murine sarcoma viral oncogene homolog B1)/MEK (mitogen-activated protein kinase)-inhibitors (BRAFi/MEKi) over the last decade has revolutionized the standard-of-care of metastatic melanoma [1,2,3,4,5,6,7,8]. Metastatic melanoma has gone through a paradigm shift from a life-threatening situation with scarce and ineffective treatment options to a disease with potent treatment options with the ability to induce prolonged survival.

The CTLA-4 (cytotoxic T lymphocyte-associated antigen 4) inhibitor ipilimumab was the first ICI to be approved by the Food and Drug Administration (FDA) and European Medicines Agency (EMA) in 2011 after showing an improvement in median overall survival (OS) of 4 months [1,2], and was primarily implemented in Denmark in 2012. After approval by regulatory authorities, the BRAFi vemurafenib was approved as monotherapy in 2011 and more widely used in Denmark during 2012, while combination therapy with the BRAFi dabrafenib and the MEKi trametinib was introduced in 2013. Treatment with anti-programmed death 1 (PD1)-inhibitor was approved and implemented in routine clinical practice in Denmark in 2016.

The approval and implementation of new drugs are followed by an increasing request for real-world evidence to support regulatory decision-making, as support tools in clinical practice, safety surveillance, generating hypotheses for further clinical testing, as well as evaluating implementation of health care services. Evidence generated from real-world data complements evidence from clinical trials in highly selected patients to a broader patient population, providing information that cannot be obtained through a clinical trial. Previous studies have shown that real-life patient cohorts may differ significantly from patients eligible for clinical trials, and evidence of benefits in real-life patients is still limited [9,10]. Therefore, in a Danish consecutively treated, nation-wide cohort of metastatic melanoma patients spanning almost a decade, we aimed to analyze OS with emphasis on long-term survivors in the pre-ICI and ICI era.

## 2. Results

A total of 1202 patients received systemic treatment in the ICI era from 2012 to 2016 compared with 552 patients in the pre-ICI era from 2008 to 2011. Median OS was significantly higher for patients starting first-line treatment in the ICI era with 11.3 months (95% confidence interval (CI) 10.3–12.3) compared with 8.3 months (95% CI 7.4–9.5, *p* < 0.0001) in the pre-ICI era (Figure 1). The corresponding 1-, 2-, 3-, 4-, and 5-year survival rates were 48%, 33%, 26%, 23%, and 21% in the ICI era compared with 36%, 18%, 13%, 11%, and 9% in the pre-ICI era, respectively. Median follow-up was 62.1 (95% CI 59.6–66.0) months.

In the ICI era, there was a higher proportion of patients with poor prognostic factors such as poor performance status (PS) and elevated LDH, while a higher proportion of patients with liver metastases was observed in the pre-ICI era (Table 1a). On the other hand, there were no differences in disease stage distribution or presence of CNS metastases between the two periods. In the pre-ICI era, 51% received high-dose interleukin-2/interferon-alpha (IL2/IFN) and 46% received chemotherapy as first-line treatment, while only a small fraction of patients received BRAFi/MEKi, anti-CTLA-4, or other first-line treatment primarily in clinical trials. First-line treatment distribution in the ICI era was more diverse as 26% received anti-CTLA-4, 20% received chemotherapy, 20% received BRAFi/MEKi, 12% received high-dose IL2/IFN, and 21% received a treatment regimen containing anti-PD1 antibodies. Survival according to subgroups is shown in Figure 2A–D.

### 2.1. Long-Term Survivors

The proportion of patients with survival above 3 years doubled from 13% in the pre-ICI era to 26% in the ICI era (*p* < 0.0001) (Table 1b). Long-term survivors in the ICI era were older and a higher proportion had received anti-PD1 at any time during their treatment course compared with long-term survivors in the pre-ICI era. There was no difference in disease stage distribution; PS; LDH level; and presence of CNS, liver, lung, or bone metastases between the two periods. It should be noted that almost no patients in PS 2 or above at baseline became long-term survivors in either treatment era, although there was a high amount of missing values. A similar tendency was not observed with elevated LDH level. Fewer long-term survivors needed additional treatment lines in the ICI era, as only 31% and 15% received third and fourth line treatment compared with 46% and 30% in the pre-ICI era (*p* = 0.0054), respectively. In other terms, a higher proportion of long-term survivors in the ICI era ended their treatment course after the second or the third treatment line compared with the pre-ICI era.

Lower disease stage at baseline was associated with a higher likelihood to become a long-term survivor both in the pre-ICI and the ICI era, as shown in Table 2. Interestingly, a higher proportion of patients with M1c became long-term survivors in the ICI era compared with the pre-ICI era (22% vs. 10%, *p* < 0.0001). Similarly, only 5% and 9% of patients that presented with liver or lung metastases became long-term survivors in the pre-ICI, which increased to 16% (*p* < 0.0001) and 23% (*p* < 0.0001) in the ICI era, respectively. For patients with bone metastases, there was no significant difference between the two periods.

### 2.2. Progression-Free Long-Term Survival

The absolute number of patients that received more than three lines of treatments was low and the following analyses were thus based on assessment of progression at the first to third line of treatment. In the pre-ICI era, the highest proportion of long-term survivors without progression was identified following first-line treatment (26%) and decreased at second-line (18%) and third-line (17%) treatment. In contrast, the proportion of long-term survivors without progression increased from first-line (35%) to second-line (44%) treatment and remained high (31%) at third-line treatment in the ICI era (Figure 3).

Long-term survivors without progression were pooled and the proportion increased significantly from 43% in the pre-ICI era to 70% in the ICI era (*p* < 0.0001). In the pre-ICI era, a third of patients without progression had received any ICI during their treatment course compared with over 90% in the ICI era, though a high proportion of patients progress on ICI in general. For the whole patient cohort, there was a significantly increase in patients without progression from 5% in the pre-ICI era to 18% in the ICI era (*p* < 0.0001).

### 2.3. Prognostic Factors for Improved Overall Survival

In univariate analyses, age below 70 years; female gender; PS of 0–1; cutaneous and mucosal melanoma type; normal LDH level; low disease stage; absence of CNS, bone, lung, and liver metastases; and ICI treatment were significantly associated with improved survival (Appendix A). Owing to the overlap between variables, disease stage and a combined ICI category were chosen together with age, gender, PS, melanoma type, and LDH to be entered in multivariate analysis using multiple imputations. Treatment with BRAFi/MEKi any time during their treatment course was not associated with survival in univariate analysis. Patients with a BRAF mutation treated with a BRAFi/MEKi had a higher proportion of poor prognostic baseline factors such as elevated LDH, poor PS, and higher disease stage compared with BRAF mutated patients who did not receive treatment with a BRAFi/MEKi any time during their treatment course (Appendix A). Fifty-two percent of patients treated with BRAFi/MEKi received it as their first-line treatment, while the majority of the non-BRAFi/MEKi treated patients received CTLA-4-inhibitor (33%) or IL2/IFN (24%) as their first-line treatment. It should be noted that some patients started their first-line treatment in the pre-ICI era and were tested later in their treatment course (data not shown).

The multivariate analysis confirmed known prognostic factors such as PS of 2 or above, elevated LDH level, and increasing disease stage together with ocular melanoma to be independently associated with decreased OS (Table 3). Treatment with ICI was significantly associated with a 61% (95% CI 0.35–0.44, *p* < 0.0001) reduction in the risk of death, irrespective of known prognostic risk factors.

## 3. Discussion

In this nation-wide retrospective analysis, we have chosen two eras; that is, 2008–2011 and 2012–2016, to reflect the change in real-life practice as a very limited number of patients received ICI before 2012. We found that median OS improved from 8.3 months in the pre-ICI era to 11.3 months in the ICI era with a corresponding increase in the tail of the curve for real-life patients. Over the years, treatment changed from ineffective chemotherapy and toxic high-dose IL2/IFN to treatments with BRAFi/MEKi and ICI. High-dose IL2/IFN can induce long-term survival for a limited patient population, while patients with poor performance, with LDH more than two times above ULN or elevated neutrophils did not benefit [11,12]. The observed increase in the number of patients treated in 2012–2016 reflects the selection of a wider patient cohort owing to better treatment options with a more tolerable toxicity profile over time supported by a higher proportion of poor prognostic features in the expanded and older patient cohort. However, the age-adjusted incidence rate in malignant melanoma over the last 10 years is increasing in Denmark [13,14] and it is uncertain if this is also followed by a corresponding increase in metastatic melanoma.

There were only minor differences in the distribution of prognostic factors between the two periods for all patients, primarily in favor of the pre-ICI era with better PS and lower LDH levels. Thus, we consider it unlikely that the improved mOS and long-term survival can be explained by these differences in baseline characteristics. It was unexpected that treatment with BRAFi/MEKi did not affect OS, as it has been shown to improve OS in phase 3 trials [3,4]. Use of BRAFi/MEKi in national practice could be biased by patient selection towards patients with symptomatic large tumor burden, rapidly progressive disease, poor prognostic factors, or patients unfit for ICI, as indicated in our data. Moreover, the results could be influenced by BRAFi given as monotherapy in the early ICI era, because MEKi was implemented in Denmark from 2013. It is important to notice that our findings do not mean that BRAFi/MEKi is not an effective treatment. Still, it should be noticed that the findings from clinical trials are not directly referable to real-life patients and selection is essential.

Nonetheless, even with an improvement in mOS to 11.3 months, metastatic melanoma still has a dismal prognosis. However, there were significant further improvements since 2015 as anti-PD1 and combination ICI became standard first-line treatment. Moreover, it is encouraging that a higher proportion of patients obtained long-term survival without progression not only at first-line treatment, but also later treatment lines, and the need for further treatment lines declined over time in the ICI era. These findings might reflect that ipilimumab initially was approved as a second-line treatment and anti-PD1 treatment became available in later lines during the implementation period. Most long-term survivors had limited disease, but also, patients with M1b and M1c disease, and there were even indications of patients with M1d disease, had a higher chance of becoming a long-term survivor in the ICI era. Still, a large proportion of real-life patients do not benefit sufficiently from the new treatment possibilities and we find it concerning that almost no patients in PS 2 become long-term survivors.

Our survival data are in coherence with data from a large hospital-based cancer-registry that covers approximately 70% of all newly diagnosed cancers in the United States, although the percentual coverage of metastatic melanoma is uncertain [15]. Their database contained comprehensive information of pre-systemic treatment with data on prior surgery, comorbidity, insurance status, LDH level, and type of first-line treatment, but not later lines. In their retrospective dataset of more than 15,000 patients with synchronous metastatic melanoma, mOS improved from 8.4 months at pre-approval (of ICI and BRAFi) to 10.2 months at post-approval for patients treated for cutaneous stage 4 melanoma. Their 4-year OS improved from 18% to 23.5%, which is similar to our 4-year OS in the ICI era, but with a lower relative increase than in our nation-wide cohort. Baseline characteristics were only reported for patients receiving ICI, while survival was reported for all patients pre- and post-approval; therefore, it is uncertain if general baseline characteristics are similar between cohorts. Mangana et al. compared survival between treatment types in a retrospective real-life cohort of 395 patients treated at three Swiss hospitals [16]. Median OS for their reference chemotherapy (treatment-naïve) group from 2008 to 2009 was 7.1 months, and comparable to our pre-ICI group, while treatment-naïve patients on targeted therapy or ICI from 2010 to 2014 had an mOS of 14.6 months, which is considerably higher than both the American cohort and our cohort. It should be noted that there was a high proportion of CNS metastases in all patient cohorts, ranging between 33% to 53%, which could be owing to meticulous screening for CNS metastases before the treatment started and not only for symptomatic CNS metastases. In both studies, a description of metastatic sites for all treated patients besides CNS metastases, PS, or subsequent treatment lines received was not reported. On the other hand, both studies had a more detailed histopathological description of the primary tumor compared with ours. These differences may make direct comparison difficult.

Many factors such as comorbidity, PS, patient involvement, and/or symptomatic metastases influence the decision of treatment strategy. It is not possible to capture these considerations in a clinical trial and the evaluation of implementation of new treatment strategies in a nation-wide cohort over time is thus essential to evaluate if survival is improved in general, as well as to examine if it is the same real-life patient-type that becomes a long-term survivor. The collaboration and organization of the Danish health care system makes it possible to gather such real-life data on an almost complete national cohort of patients with metastatic melanoma, with the limitation being that missing values for LDH and PS and the lack of BRAF mutation status in the pre-ICI era might bias data.

## 4. Materials and Methods

### 4.1. Patient Population

All patients treated consecutively for metastatic melanoma between 1 January 2008 and 31 December 2016 were included in this retrospective study. Records were retrieved from the Danish Metastatic Melanoma Database (DAMMED). The database is nation-wide and includes data on baseline characteristics, disease stage, treatment lines received, response to treatment, reasons for treatment discontinuation, date of progression, and time of death. The database has an estimated coverage of >95% of all metastatic melanoma diagnoses within Denmark. Treatment is centralized at the departments of Oncology at Herlev, Odense, Aalborg, and Aarhus University Hospital.

The study was approved by the Danish Patient Safety Authority (3–3013-2306/1) and the Data Protection Agency (2012–58-0004).

### 4.2. Treatment and Response

Patients were treated according to best practice at time of referral with chemotherapy (primarily temozolomide), immunotherapy (combined high-dose interleukin-2 and interferon-alpha (IL2/IFN)), immune checkpoint-inhibitors (ipilimumab, pembrolizumab, nivolumab), BRAFi (vemurafenib, dabrafenib, encorafenib) alone or in combination with MEKi (cobimetinib, trametinib, binimetinib), or included in a clinical trial. Patients were analyzed according to treatment type, time-period, and survival below or above 3 years. Ipilimumab was approved in 2011 (EMA and FDA), however, the cut-off between the pre-ICI era (2008–2011) and the ICI era (2012–2016) was chosen as ipilimumab was generally implemented in the clinic in Denmark in 2012.

Tumor response was assessed according to the Response Evaluation Criteria in Solid Tumor (RECIST) guidelines 1.0 [17]. Progression-free survival was defined as the time from initiation of systemic treatment to the date of documented disease progression or censored at last follow-up (31 December 2019). OS was defined as the time from initiation of first-line systemic treatment to death or censored at last follow-up.

### 4.3. Statistical Analyses

Differences in distributions of baseline characteristics were evaluated with the chi-square test. Median OS was analyzed with the Kaplan–Meier method and compared with the log-rank test. A *p*-value < 0.05 was considered statistically significant. Median follow-up time was analyzed with the reverse Kaplan–Meier method.

Univariate analysis was performed with the log-rank test and a significance level of 0.05 was chosen as an entry for the multivariate analysis using the Cox proportional hazard model with reported hazard ratios (HRs) and corresponding 95% confidence intervals (CIs). Missing baseline values were imputed using multiple imputations with the chained equation method under the missing-at-random assumption.

Statistical analyses were performed with R [18].

## 5. Conclusions

The implementation of ICI has led to a significant increase in progression-free, long-term survival for real-life patients with metastatic melanoma.

## Figures and Tables

**Figure 1 cancers-12-02591-f001:**
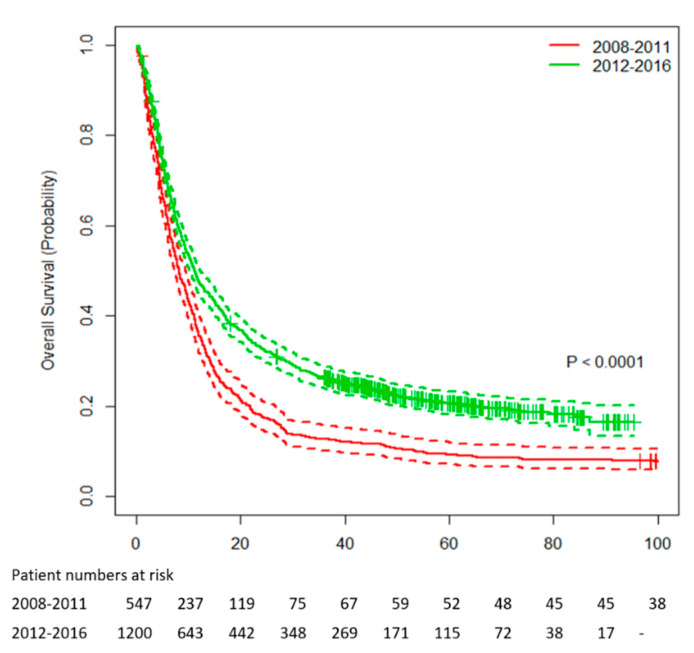
Overall survival for patients with metastatic melanoma between 2008–2011 and 2012–2016.

**Figure 2 cancers-12-02591-f002:**
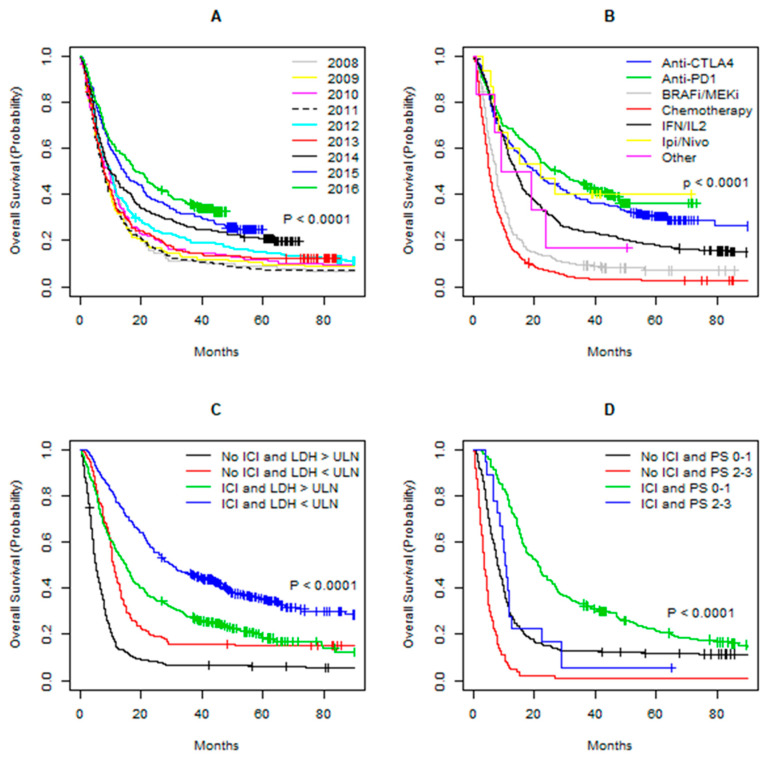
Kaplan–Meier curves of overall survival (OS) for patients with metastatic melanoma. (**A**) OS per treatment year. (**B**) OS per type of first-line treatment. (**C**) OS for patients with LDH above or below the upper level of normal (ULN) with or without immune checkpoint-inhibitor (ICI) therapy any time during their treatment course. (**D**) OS for patients in performance status (PS) 0–1 or 2–3 with or without ICI therapy any time during their treatment course. Ipi/Nivo = Ipilimumab/Nivolumab; IL2/IFN = interleukin-2/interferon-alpha.

**Figure 3 cancers-12-02591-f003:**
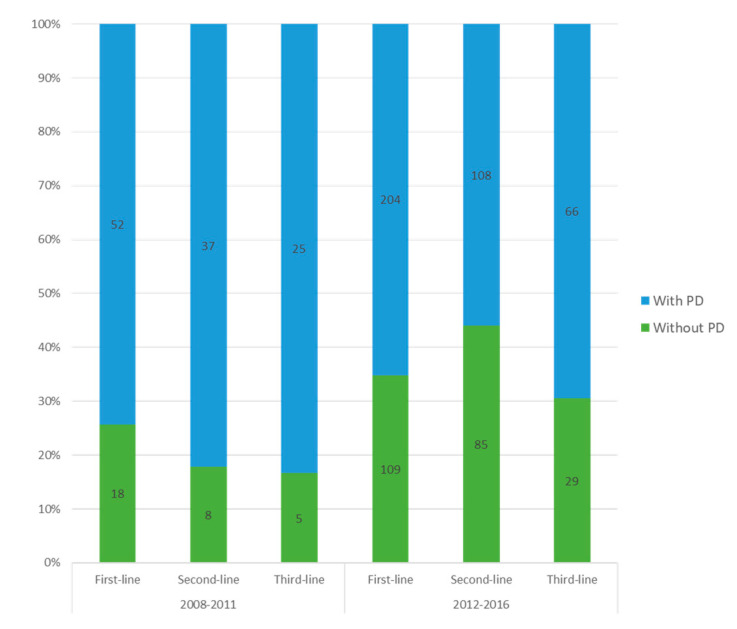
Histogram of long-term survivors with or without progression on first-, second-, and third-line treatment. Number of patients is indicated on the bars; PD = progressive disease.

**Table 1 cancers-12-02591-t001:** Baseline characteristics between 2008–2011 and 2012–2016.

	a. All Patients			b. Long-Term Survivors ^a^		
Variable	2008–2011	2012–2016	*p* Value ^b^	2008–2011	2012–2016	*p* Value ^b^
Total number of patients	552	1202		70 (13)	313 (26)	<0.0001
Emigrated (%)	3 (<1)	2 (<1)		0	0	
Age			<0.0001			0.0007
<70 years	410 (74)	739 (62)		60 (86)	203 (65)	
≥70 years	141 (26)	463 (39)		10 (14)	110 (35)	
Male gender (%)	336 (61)	687 (57)	0.1428	36 (51)	116 (55)	0.659
Melanoma type			0.1443			0.1267
Cutaneous	420 (76)	928 (77)		59 (84)	257 (82)	
Mucosal	18 (3)	52 (4)		0 (0)	15 (5)	
Ocular	30 (6)	81 (7)		0 (0)	6 (2)	
Unknown primary	82 (15)	139 (12)		11 (16)	35 (11)	
NA	2	2		0	0	
PS (%)			<0.0001			0.9151
0–1	275 (92)	325 (76)		61 (98)	71 (99)	
2–3	25 (8)	102 (24)		1 (2)	1 (1)	
NA	252	775		8	241	
BRAF status (%)	^c^			^c^		
Wildtype		546 (50)			162 (53)	
Any mutation		551 (50)			142 (47)	
NA		105			8	
LDH (%)			0.0243			0.736
Below ULN	153 (54)	431 (46)		38 (63)	187 (66)	
Above ULN	133 (47)	508 (54)		22 (37)	94 (34)	
NA	266	263		10	28	
Stage (%)			0.2682			0.9146
M1a	65 (12)	173 (15)		20 (29)	89 (29)	
M1b	81 (15)	145 (12)		16 (23)	59 (19)	
M1c	294 (54)	624 (53)		29 (41)	134 (44)	
M1d	107 (20)	226 (19)		5 (7)	25 (8)	
NA	5	34		0	5	
Metastatic sites						
CNS (%)	107 (20)	226 (19)	0.7312	5 (7)	25 (8)	0.8121
NA	5	4		0	0	
Bone (%)	120 (22)	293 (25)	0.1278	11 (16)	41 (13)	0.6055
NA	5	45		0	6	
Liver (%)	216 (40)	388 (34)	0.0165	10 (14)	61 (20)	0.2809
NA	5	45		0	6	
Lung (%)	299 (55)	580 (50)	0.0805	28 (40)	135 (44)	0.5448
NA	5	45		0	6	
First line treatment (%)			<0.0001			<0.0001
IL2/IFN	283 (51)	138 (12)		61 (87)	40 (13)	
Chemotherapy	255 (46)	241 (20)		7 (10)	11 (4)	
Anti-PD1	0 (0)	252 (21)		0 (0)	111 (36)	
Anti-CTLA-4	7 (1)	306 (26)		1 (1)	116 (37)	
BRAFi/MEKi	4 (1)	239 (20)		1 (1)	22 (7)	
ICI (blinded)	0	3 (<1)		0 (0)	2 (1)	
Dabratram/Trametinib/Pembrolizumab	0	5 (<1)		0 (0)	4 (1)	
Ipilimumab/Nivolumab	0 (0)	15 (1)		0 (0)	6 (2)	
Other ^d^	3 (1)	3 (<1)		0 (0)	1 (<1)	
Anti-CTLA-4 during 1.-7. line of therapy			<0.0001			0.4287
Yes	131 (24)	561 (47)		39 (57)	193 (62)	
No	419 (76)	639 (53)		30 (44)	120 (38)	
NA	2	3		1	0	
Anti-PD1 during 1.-7. line of therapy			<0.0001			<0.0001
Yes	18 (3)	529 (44)		16 (24)	241 (77)	
No	529 (97)	669 (56)		52 (77)	72 (23)	
NA	5	4		2	0	
BRAFi/MEKi during 1.-7. line of therapy			<0.0001			0.5326
Yes	46 (8)	434 (36)		19 (28)	75 (24)	
No	503 (92)	767 (64)		50 (73)	238 (76)	
NA	3	1		1	0	
Any anti-CTLA-4 or anti-PD1 during 1.-7. line of therapy			<0.0001			<0.0001
Yes	133 (24)	790 (66)		41 (59)	290 (93)	
No	417 (76)	411 (34)		28 (41)	23 (7)	
NA	2	1		1	0	
Best objective response (%)			<0.0001			0.5224
CR	23 (8)	105 (11)		22 (35)	97 (33)	
PR	47 (16)	242 (26)		18 (29)	81 (28)	
SD	108 (36)	237 (25)		17 (27)	66 (23)	
PD	126 (41)	365 (39)		6 (10)	49 (17)	
NA	248	253		7	20	
Treatment lines received (%)			0.0004			0.0054
1	552 (100)	1202 (100)		70 (100)	313 (100)	
2	206 (37)	614 (51)		47 (67)	193 (62)	
3	79 (14)	281 (23)		32 (46)	96 (31)	
4	30 (5)	109 (9)		21 (30)	46 (15)	
5	20 (4)	47 (4)		19 (27)	29 (9)	
6	5 (1)	16 (1)		5 (7)	12 (4)	
7	0 (0)	4 (<1)		0 (0)	4 (1)	

^a^ Overall survival above 3 years; ^b^ chi-square-test; ^c^ analysis of BRAF mutation status was not performed; ^d^ dendritic cell vaccination, T cell therapy, intralesional IL2; PS = performance status; NA = not applicable; IL2/IFN = interleukin-2/interferon-alpha; ICI = immune checkpoint-inhibitor; CR = complete response; PR = partial response; SD = stable disease; PD = progressive disease; ULN = upper level of normal.

**Table 2 cancers-12-02591-t002:** Disease stage and metastatic sites according to survival status between 2008–2011 and 2012–2016.

	2008–2011		2012–2016		
	Pts with OS < 3 Years (%)	Pts with OS ≥ 3 Years (%)	Pts with OS < 3 Years (%)	Pts with OS ≥ 3 Years (%)	*p* Value ^a^
Melanoma stage					
M1a	44 (69)	20 (31)	84 (49)	89 (51)	0.0056
M1b	65 (80)	16 (20)	85 (59)	59 (41)	0.0012
M1c	264 (90)	29 (10)	490 (79)	134 (22)	<0.0001
M1d	100 (95)	5 (5)	201 (89)	25 (11)	0.0632
Metastatic sites					
Bone	108 (91)	11 (9)	252 (86)	41 (14)	0.1883
Liver	205 (95)	10 (5)	327 (84)	61 (16)	<0.0001
Lung	269 (91)	28 (9)	444 (77)	135 (23)	<0.0001

^a^ Pearson’s chi-square test. Pts = patients; OS = overall survival.

**Table 3 cancers-12-02591-t003:** Multivariate analysis with multiple imputations 2008–2016.

	HR ^a^	95% CI	*p* Value
Age			0.3876
<70 years	1.00		
≥70 years	1.05	0.93–1.19	
Gender			0.0886
Female	1.00		
Male	1.10	0.98–1.23	
PS			<0.0001
0–1	1.00		
≥2	1.83	1.39–2.40	
LDH			<0.0001
Below ULN	1.00		
Above ULN	1.91	1.77–2.12	
Stage			
M1a	1.00		
M1b	1.27	1.01–1.61	0.0385
M1c	1.79	1.49–2.16	<0.0001
M1d	2.64	2.14–3.28	<0.0001
Melanoma type			
Cutaneous	1.00		
Mucosal	1.20	0.90–1.61	0.2071
Ocular	1.57	1.18–2.09	0.0016
Unknown primary	1.02	0.86–1.21	0.8156
Any anti-CTLA-4 or anti-PD1 therapy ^b^			<0.0001
No	1.00		
Yes	0.39	0.35–0.44	

^a^ Cox regression analysis, *p* < 0.05 in univariate analyses as entry; ^b^ anytime during their treatment course. HR = hazard ratio; CI = confidence interval; PS = performance status; ULN = upper level of normal.

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
