# Peer review of "Improved Progression-Free Long-Term Survival of a Nation-Wide Patient Population with Metastatic Melanoma"

_cancers, 2020, doi:10.3390/cancers12092591_

Round 1

Reviewer 1 Report

In the manuscript, the authors extracted clinical data on all patients who received systemic treatment for metastatic melanoma between 2008 and 2016 from the Danish Metastatic Melanoma database, and compared the overall survival and long-term survival before and after the application of the first approved immune checkpoint-inhibitor (ICI) Ipilimumab in 2012. With clear clinical evidence, the authors showed that implementation of ICI has led to a significant increase in progression-free, long-term survival for real-life patients with metastatic melanoma. These findings are pretty solid and the design is straightforward, and the study can provide the readers some useful instant knowledge about the treatment effects of ICI in real world.

I only have some minor concerns regarding the introduction part. The introduction is too succinct and should be extended more. The authors can consider to talk a little bit more about the epidemic of melanoma around the world and in Denmark. Some more background information about the Ipilimumab (and other ICIs used in clinical) would also be helpful.

Author Response

Dear Reviewer

We would like to thank you for your review and comment. Please see our answers below. We hope that you will find the revision satisfiable.

Point 1: I only have some minor concerns regarding the introduction part. The introduction is too succinct and should be extended more. The authors can consider to talk a little bit more about the epidemic of melanoma around the world and in Denmark. Some more background information about the Ipilimumab (and other ICIs used in clinical) would also be helpful.

Response 1: We agree that the introduction is too brief and has been extended in the revised manuscript with more details on treatment used in the clinic.

Kind regards

The authors

Reviewer 2 Report

The authors have reviewed the DAMMED to assess the overall survival of patients with stage IV melanoma treated in the years before introduction of immune checkpoint inhibitors in comparison to patients treated after the introduction. A strength of this study is the large number of individuals and the long follow-up (5 years). They found that the overall survival was significantly higher in patients treated with ICI compared to patients treated in the pre-ICI era. However there are some weaknesses in the paper:

  • In the introduction the authors mention that this study will provide results from "trial-excluded"patients. However, in the pre-ICI era 2 patients were treated with BRAF MEK and anti-CTLA4, I presume in a trial. Evenso for the patients treated with triple therapy and blinded ICI in the post-ICI era. These patients should have been excluded.
  • the sum of the numbers in table 1 and supplementary table S1 are not equal, even not for the variables without missing values. EG Age < 70 years: 410+ 739 = 1149 in table 1 and in S1 the N=1144. I wonder if the authors can explain the differences?
  • In sentence 123 gender cannot be associated with improved survival, what is meant is female gender
  • In sentence 127 the authors forgot to mention "gender" since it is in the table as a variable.
  • the first paragraph should mention your main results. And thus the first paragraph should mention the OVERALL SURVIVAL difference found
  • there is little information about the BRAF MEKi treatment. When was braf meki approved in Denmark? In what line were patients treated? In the discussion some possible explanations for the fact that there was no overall survival benefit for treatment with BRAF MEKi were mentioned. It would certainly be of strenth ifthe authors could provide some of the data. Is it the case that patients treated with braf mek inhibitors had larger tumour burden, more poor prognostic factors etc?
  • By  chosing PFS and OS to start from the moment of start of systemic therapy instead of diagnosis of metastatic disease or first visit to the oncology, there might be a bias favouring patients with rapid progressive disease. Patients with small volume disease can have some waiting time before starting therapy. Why did you chose this way?

Author Response

Dear Reviewer 2

We would like to thank you for your review and comments. We hope that you will find our answers satisfying.

The authors have reviewed the DAMMED to assess the overall survival of patients with stage IV melanoma treated in the years before introduction of immune checkpoint inhibitors in comparison to patients treated after the introduction. A strength of this study is the large number of individuals and the long follow-up (5 years). They found that the overall survival was significantly higher in patients treated with ICI compared to patients treated in the pre-ICI era. However there are some weaknesses in the paper:

Point 1: In the introduction the authors mention that this study will provide results from "trial-excluded"patients. However, in the pre-ICI era 2 patients were treated with BRAF MEK and anti-CTLA4, I presume in a trial. Evenso for the patients treated with triple therapy and blinded ICI in the post-ICI era. These patients should have been excluded.

Response 1: We understand that our sentence on line 58-59 (revised manuscript) in the introduction section can be misunderstood as if our cohort consist of trial-excluded patients. However, this was a reference to another publication analyzing “trial-excluded” patients. Our cohort consist of all patients treated consecutively between 2008 and 2016 as mentioned in the method section. This has been clarified in the introduction (line 58-59) and on line 238 in the methods section. The described trial patients remain included in the analyses to ensure a complete national cohort and not a selected patient cohort.

Point 2: the sum of the numbers in table 1 and supplementary table S1 are not equal, even not for the variables without missing values. EG Age < 70 years: 410+ 739 = 1149 in table 1 and in S1 the N=1144. I wonder if the authors can explain the differences?

Response 2: The database is almost complete for age. However, to perform the univariate analyses a date of death or censoring date is also required. This was lacking or inaccurate in some patients which leads to the described differences. A note has been added to the supplementary table S1.

Point 3: In sentence 123 gender cannot be associated with improved survival, what is meant is female gender

Response 3: We agree this should be specified more clearly and “female” has been added to line 147 in the revised manuscript.

Point 4: In sentence 127 the authors forgot to mention "gender" since it is in the table as a variable.

Response 4: Gender has been added to the sentence on line 151 in the revised manuscript.

Point 5: the first paragraph should mention your main results. And thus the first paragraph should mention the OVERALL SURVIVAL difference found

Response 5: We presume that the comment refers to the first paragraph in the results section. The overall survival results have been added to the first paragraph on lines 64-68, and figure 1 and 2 moved accordingly. Similarly, the sentence concerning survival has been added to the first paragraph on lines 170-172 in the discussion section.

Point 6: there is little information about the BRAF MEKi treatment. When was braf meki approved in Denmark? In what line were patients treated? In the discussion some possible explanations for the fact that there was no overall survival benefit for treatment with BRAF MEKi were mentioned. It would certainly be of strenth ifthe authors could provide some of the data. Is it the case that patients treated with braf mek inhibitors had larger tumour burden, more poor prognostic factors etc?

Response 6: Patients were not tested for BRAF mutation in the pre-ICI and information was therefore omitted to focus on the 2 time periods. It is understandable that more information is wanted as it is surprising that treatment with BRAFi/MEKi was not associated with survival. Information on BRAFi/MEKi treatment has therefore been added in the revised manuscript on lines 45-51 in the introduction section, on lines 153-160 in the result section, on lines 190-192 in the discussion together with an added supplementary table S2.

Point 7: By  chosing PFS and OS to start from the moment of start of systemic therapy instead of diagnosis of metastatic disease or first visit to the oncology, there might be a bias favouring patients with rapid progressive disease. Patients with small volume disease can have some waiting time before starting therapy. Why did you chose this way?

Response 7: Data from diagnose of metastatic disease or first visit would be desirable to give more detailed information. However, only the 1st treatment day is consequently registered in the DAMMED database. Also, this date is registered in clinical trials.

Kind regards

The authors

Round 2

Reviewer 2 Report

no further comments.